# Intranasal Treatment of Ferrets with Inert Bacterial Spores Reduces Disease Caused by a Challenging H7N9 Avian Influenza Virus

**DOI:** 10.3390/vaccines10091559

**Published:** 2022-09-19

**Authors:** Joe James, Stephanie M. Meyer, Huynh A. Hong, Chau Dang, Ho T. Y. Linh, William Ferreira, Paidamoyo M. Katsande, Linh Vo, Daniel Hynes, William Love, Ashley C. Banyard, Simon M. Cutting

**Affiliations:** 1Animal and Plant Health Agency (APHA), Woodham Lane, Weybridge KT15 3NB, Surrey, UK; 2SporeGen Ltd., London Bioscience Innovation Centre, 2 Royal College Street, London NW1 0NH, UK; 3HURO Biotech JSC, Lot A1-8, VL3 Road, Vinh Loc 2 Industrial Park, Long Hiep Commune, Ben Luc District, Long An, Vietnam; 4Department of Biological Sciences, Royal Holloway University of London, Egham TW20 0EX, Surrey, UK; 5Destiny Pharma Plc., Sussex Innovation Centre, Science Park Square, Falmer, Brighton BN1 9SB, UK

**Keywords:** innate immunity, influenza, bacterial spores, prophylaxis, *Bacillus subtilis*

## Abstract

*Background:* Influenza is a respiratory infection that continues to present a major threat to human health, with ~500,000 deaths/year. Continued circulation of epidemic subtypes in humans and animals potentially increases the risk of future pandemics. Vaccination has failed to halt the evolution of this virus and next-generation prophylactic approaches are under development. Naked, “heat inactivated”, or inert bacterial spores have been shown to protect against influenza in murine models. *Methods:* Ferrets were administered intranasal doses of inert bacterial spores (DSM 32444^K^) every 7 days for 4 weeks. Seven days after the last dose, the animals were challenged with avian H7N9 influenza A virus. Clinical signs of infection and viral shedding were monitored. *Results:* Clinical symptoms of infection were significantly reduced in animals dosed with DSM 32444^K^. The temporal kinetics of viral shedding was reduced but not prevented. *Conclusion:* Taken together, nasal dosing using heat-stable spores could provide a useful approach for influenza prophylaxis in both humans and animals.

## 1. Introduction

COVID-19 has refocused attention on potential solutions not only to the current pandemic but also to future threats. One candidate that might be considered is the use of bacterial spores. Spores of *Bacillus subtilis* have been used extensively as vaccine delivery vehicles, either by engineering spores to express antigens or by adsorption of heterologous antigens to the spore surface [1,2]. With the adsorption approach, it has been shown that heat-inactivated H5N1 influenza virions (NIBRG-14 clade) delivered by loaded spores conferred localized immunity as well as protection in murine models of infection [3]. Cross neutralization of other clades as well as antigen sparing were also demonstrated using this approach [3]. Remarkably, inactivated (inert) spores carrying no antigen (naked spores), administered intranasally, also conferred protection to a mouse-adapted strain of H5N2 [3]. The underlying mechanism was shown to be innate immunity via TLR (Toll-like receptor)-mediated expression of NF-κβ and recruitment of NK (Natural Killer) cells into the lungs together with the maturation of DCs (Dendritic cells) [3,4]. Spores have a number of attributes that may directly or indirectly be involved in this phenomenon. First, they have been shown to activate TLRs [3,5,6]. Second, they have adjuvant properties, and when co-administered with antigens, whether by the same (mucosal) or different routes (systemic antigen—mucosal spores), they induce localized immunity as well as direct balanced antigen-specific Th1-Th2 immune responses [7].

Seasonal A and B viruses circulate among humans, with resulting epidemics of acute respiratory disease estimated at 3–5 million cases/year and up to 650,000 mortalities/year [8,9]. Infections resulting from zoonotic influenza A viruses can cause severe illness and contribute to the emergence of pandemic strains. Current and recent outbreaks of avian and swine influenza not only afflict farmed animals but also increase the risk of transfer to humans and potentially increase the risk of a future pandemic [10]. It seems appropriate then to evaluate whether the use of inert spores might protect against influenza in a robust model of influenza infection. The ferret, guinea pig, and Syrian hamster models of infection are superior to the murine models since they allow assessment of not only infection but also transmission [11,12,13,14].

Novel human influenza A virus (IAV) infections during the past decade have included the H7N9 subtype, first isolated from humans in China in early 2013 [15]. In this paper, we investigate the protective efficacy of inert *B. subtilis* spores in the ferret model of influenza infection, using the H7N9 subtype for challenge. We show that intranasal applications of inert spores showed a significant reduction in clinical signs of disease but did not substantially impact viral shedding.

## 2. Materials and Methods

### 2.1. Spore Inoculum

The *B. subtilis* strain DSM 32444 was used for this study. This strain is recommended as safe for human consumption (USA FDA GRAS-notification GRN 000905). Spores were prepared in batch culture (800 mL), and suspensions in dH_2_O were inactivated by autoclaving and are henceforth referred to as DSM 32444^K^. A validation of spore inactivation was made by serial dilution of heat-treated spore suspensions and plating for viability on an agar growth medium and with no resulting bacterial growth, meeting the required standard. The analysis was conducted in triplicate using standard operating procedures. Note that autoclaved spores did not rupture and retained their refractility, shape, and size [16,17]. Aliquots (1 mL) were used for animal studies, with each lot containing 5 × 10^10^ inactivated spore particles (determined by microscopic counting using a Neubauer counting chamber). All sample analyses were conducted using validated methods, and all manufacturing was conducted in compliance with current Good Manufacturing Practice (WHO GMP) at HURO Biotech (Long An, Vietnam).

### 2.2. Virus

A zoonotic avian influenza virus strain A/Anhui/1/13 [H7N9] (accession no. EPI4399507, EPI439509) was used for the infectious challenge [18].

### 2.3. Ferret Study

United Kingdom regulations categorize H7N9 as a Specified Animal Pathogens Order (SAPO) 4 and Advisory Committee on Dangerous Pathogens (ACDP) Hazard Group 3 pathogen because it is a notifiable animal disease agent and presents a zoonotic risk; hence, the stages of the *in vivo* experiment involving the H7N9 virus were conducted in licensed containment level 3 facilities at the Animal and Plant Health Agency (APHA), Weybridge, Surrey, UK. Two groups of ferrets (*Mustela putorius furo*; non-albino) each consisting of 8 animals (4 male, 4 female; ~750–1000 g) were acclimatized for 7 days prior to treatment. Immediately after this period (Day 0), the animals were considered free of current influenza infection with HIT titres ≤1/5 and absence of viral RNA in nasal wash samples. The animals were housed in groups (*n* = 4) by sex in open cages. At Days 0, 7, 14, and 21, the animals were intranasally administered 0.5 mL of DSM 32444^K^ (Group 1) or PBS (Group 2) (0.25 mL/nare). For DSM 32444^K^, this equated to a total of 2.5 × 10^10^ spore particles per dose. On Day 28, the animals were transferred to BSL3 facilities and all were challenged intranasally (0.5 mL/nostril) with AIV (A/Anhui/1/13 [H7N9]) at a titre of 1 × 10^7^ TCID_50_/mL (tissue culture infectious dose). The viral titre was back-titrated following inoculation by TCID_50_ to ensure that an accurate viral titre had been administrated. Note that the H7N9 challenge dose for ferrets is consistent with that used by others [19].

Weight and temperatures were taken daily from all ferrets. Nasal washes were performed every other day post-infection, and detection and quantification of viral RNA were performed by RT-qPCR. Fifteen days post-infection (d.p.i.), the animals were euthanized and blood was taken for serological analysis (with the exception of one animal that was culled prematurely).

### 2.4. Assessment of Clinical Scores

All ferrets were assessed daily for clinical symptoms using a standardized scoring system (Appendix A). Weights and temperatures were taken at daily intervals, and body temperatures were monitored using subdermal microchips implanted in the nape of the ferret.

### 2.5. Sampling and Analysis

For nasal washes, ferrets were anaesthetized using isoflurane (IsoFlo, Zoetis) and 1 mL (0.5 mL/nare) of Dulbecco’s PBS (Sigma, St. Louis, MO, USA) was administered intranasally and used to lavage the nasal cavity as described previously [18]. The nasal washes were collected into 2 mL Eppendorf tubes and stored at −80 °C until required. Blood was obtained both prior to infection via venipuncture and post-infection via cardiac puncture under terminal anaesthesia. Blood was stored at 4 °C to facilitate clotting and centrifuged at 1000× *g* for 15 min, and the serum was aliquoted into fresh tubes.

### 2.6. Serology

The serum was treated with a receptor-destroying enzyme (RDE) (supplied and validated by APHA) and heat-treated at 56 °C for 30 min; seroconversion to AIV H7N9 was assessed using the hemagglutination inhibition assay, as previously described [18]. Beta-propiolactone (BPL) (97%, Sigma) inactivated homologous A/Anhui/1/13 [H7N9] antigen [20] was used to quantify hemagglutination inhibition titres (HITs). HITs ≤ 1/5 were considered negative.

### 2.7. RT-qPCR Analysis

RNA was extracted from ferret nasal wash samples using the KingFisher™ Flex Purification System and MagMax™-96 Total RNA Isolation Kit technology (Invitrogen) and eluted in 90 μL of DEPC-treated RNase free H_2_O. Influenza A virus RNA was detected using an influenza A-specific RT-qPCR targeting the Matrix-gene, as described elsewhere [21]. The virus titre was determined by using a serially diluted standard curve generated from RNA extracted from the homologous virus (A/Anhui/1/13 [H7N9]) at a starting titre of 1 × 10^7^ TCID_50_/mL, as previously described [18]. The final viral titre in each sample was presented as relative equivalency units (REUs), calculated based on Cq (quantification cycle) values, and extrapolated from the standard curve. The limit of detection was a value of ≥36.00 Cq (≥7.012 REUs) based on the threshold for influenza A virus diagnostic testing [21].

## 3. Results

### 3.1. Symptomatic Protection to H7N9 AIV Infection

Prior to the challenge, the animals administered four intranasal doses of DSM 32444^K^ or PBS (control group) exhibited no clinical signs including weight loss or elevated temperatures (Appendix A). After the last dose of DSM 32444^K^ or PBS, the animals were infected (intranasally) with H7N9. One ferret (no. 323) in the control group was euthanized at 7 days post-infection (d.p.i.). on welfare grounds due to prolonged weight loss, loss of appetite, and dehydration.

For the remaining animals, clinical signs were monitored and graded (Appendix A). Infection with the H7N9 virus typically induces a range of clinical signs including lethargy, loss of appetite, change in fur appearance, dehydration, and elevated temperature. The animals in the control group exhibited more severe clinical signs compared to the animals receiving DSM 32444^K^ spores (Figure 1). All ferrets exhibited mild and transient pyrexia, starting from 1 d.p.i., with a maximum of a 2 °C increase from the predefined baseline (Figure 2A). Elevated temperatures returned to baseline levels from 4 to 6 d.p.i. for all groups. Pyrexia was most pronounced in the control-treated female group, and at 2 d.p.i., the DSM 32444^K^-treated animals exhibited a significantly (*p* < 0.05) lower body temperature. Most ferrets exhibited weight loss, with a 20% reduction in weight from baseline data between 1 and 5 d.p.i. (Figure 2B). The control-treated groups exhibited significantly (*p* < 0.0001) greater weight loss compared to DSM 32444^K^-treated ferrets (Figure 2B). Taken together, this study shows that intranasal administration of DSM 32444^K^ spores reduced the symptoms of infection caused by a zoonotic influenza A virus.

### 3.2. Protection against Viral Shedding

All ferrets shed detectable viral H7N9 RNA between 2 × 10^8^ and 2 × 10^4^ REU.mL^−1^ (REU, relative equivalency units) from their nasal cavities at 2 d.p.i., indicating that all animals had become productively infected with H7N9 (Figure 3). Shedding was detectable in all ferrets, above the limit of detection, until 10 d.p.i. for the control group and 6 d.p.i. for the DSM 32444^K^-treated group (Figure 3). However, there was no statistical difference in viral shedding between groups using one-way ANOVA at any time point following infection. These data suggest that prior treatment with DSM 32444^K^ does not alter the titre of the virus being shed or overall virus load following infection with avian influenza A.

The infectivity of the shed virus was not determined, and although rRT-qPCR is highly sensitive and was used here to detect viral RNA, it is unable to provide insight into the titres of infectious virus present within the sample for which egg isolation is traditionally used. Despite this limitation, it appears that dosing with spores has little if any impact on viral shedding. This indicates that protection most likely occurs through an unspecific mechanism such as innate immunity. Here, spore-induced innate immunity is unable to provide sufficient protection to impair transmission but it is possible that further refinement of dose and/or dosing regimens may achieve a measurable impact on transmission. Current licensed systemic influenza vaccines provide, at best, 60% protection and limited effect on transmission. Accordingly, killed spores might have utility as an adjunct to existing flu vaccines.

### 3.3. Effect of DSM 32444^K^ on Seroconversion to Influenza A Virus

Serum was collected from all ferrets at 15 d.p.i. (with the exception of the one euthanized animal). All ferret serum was treated with RDE and tested via the haemagglutinin inhibition assay using homologous H7N9 antigen. All ferrets exhibited seroconversion to the H7N9 antigen (HIT titres > 29), confirming active infection.

## 4. Discussion

This study strengthens an earlier finding [3] that intranasal dosing of mice with inert *B. subtilis* spores has the ability to reduce symptoms of influenza. Here, we used the ferret model of influenza, which is generally considered more informative since it closely mimics infection in humans presenting both the symptoms of the disease as well as replication and shedding of the virus. Our data show that pre-treatment with inert spores of *B. subtilis* leads to a clear reduction in the virulent and symptomatic stages of influenza virus infection. However, the shedding of the virus was not significantly affected. The reduction in clinical symptoms in the more robust ferret model is encouraging. Firstly, it agrees with our earlier mouse study (which measured only clinical signs and symptoms rather than viral shedding), and secondly, it shows that inert spores have potential utility for controlling this viral disease in humans and/or in animals. It seems likely that, with further refinement and modification of either dose or dosing regimen, increased levels of protection from disease might be achieved. As reported previously [3], the underlying mechanism is most probably that of innate immunity.

Innate immunity plays an important role in influenza [22]. One of the key mechanisms that evoke innate immunity is the interaction of heterologous ligands with pattern recognition receptors, typically TLRs, displayed on the surface of host cells. Spores have been shown to interact with TLR2, 4, and 6, and studies using TLR2 knockout mice have shown the interaction of spores with TLR2 of DCs and direct induction of the MyD88 signalling pathway [4], resulting in DC maturation and induction of adaptive responses. A number of TLR agonists have been shown to have efficacy in preventing influenza disease [23,24,25,26]. Inert spores then add to the number of TLR agonists that are able to prevent the symptoms of influenza. *In vitro* studies have shown that spores are phagocytosed and able to persist within the phagosome significantly longer than vegetative *Bacillus* cells [2,27]. This persistence is likely important and may mimic, in part, the behaviour of an intracellular pathogen [27]. In murine studies, the progression of spores applied intranasally has been examined in detail [2]. Haematoxylin–eosin staining revealed spores in the bronchioles after 2 h and by 6 h infiltration to the alveoli. Phagocytosis in lung cells was observed, but after 24 h, spores were no longer detectable, suggesting that phagocytosis had cleared them. The size of the spores (~1 μm) is consistent with the size range (1–5 μm) suitable for phagocytic uptake and transit across M (microfold) cells [28].

Although less is known about the underlying mechanisms of innate immunity in ferrets [29,30], it is generally understood that the host’s innate immunity plays an important role in disease and resolving infection [22]. One aspect of this is the production of pro-inflammatory cytokines in the upper respiratory tract (URT) and the production of sIgA [29,31]. In humans, nasal administration of spores has been shown to induce a number of cytokines, stimulating the production of sIgA [32] as well as inducing a Th1 bias [33]. We would predict that the underlying mechanisms (interaction with TLRs, cytokine induction, recruitment of NK cells, and maturation of DCs) inhibiting influenza infection are broadly similar in mice and ferrets.

In this case though, the use of spores has a number of unique attributes that lend themselves to further development. Bacterial spores are produced simply and cost-effectively using bioreactors, and spores are commercially produced as probiotics or animal feed products [34]. The majority of *Bacillus* strains that are used are considered safe for human consumption [35], and humans and animals are exposed to these ubiquitous and environmentally acquired bacteria on a daily basis, with exposure estimated at 10^4^–10^5^ spores/day [36].

## 5. Conclusions

Inert bacterial spores have been shown to have efficacy in reducing the symptoms of influenza in a robust animal model. Influenza is a disease that has a global impact on both humans and animals. Considering that pandemics are likely to occur in the future, it seems prudent to consider the use of inert spores as a potentially low-cost and pragmatic prophylactic measure for the control of influenza infections in animals and possibly also in humans. Most probably, this would be as an adjunct measure to existing flu vaccines. Since the underlying mechanism of spore application is innate immunity, it will be of interest to know whether inert spores might have a broader spectrum of efficacy against other viral pathogens.

## Figures and Tables

**Figure 1 vaccines-10-01559-f001:**
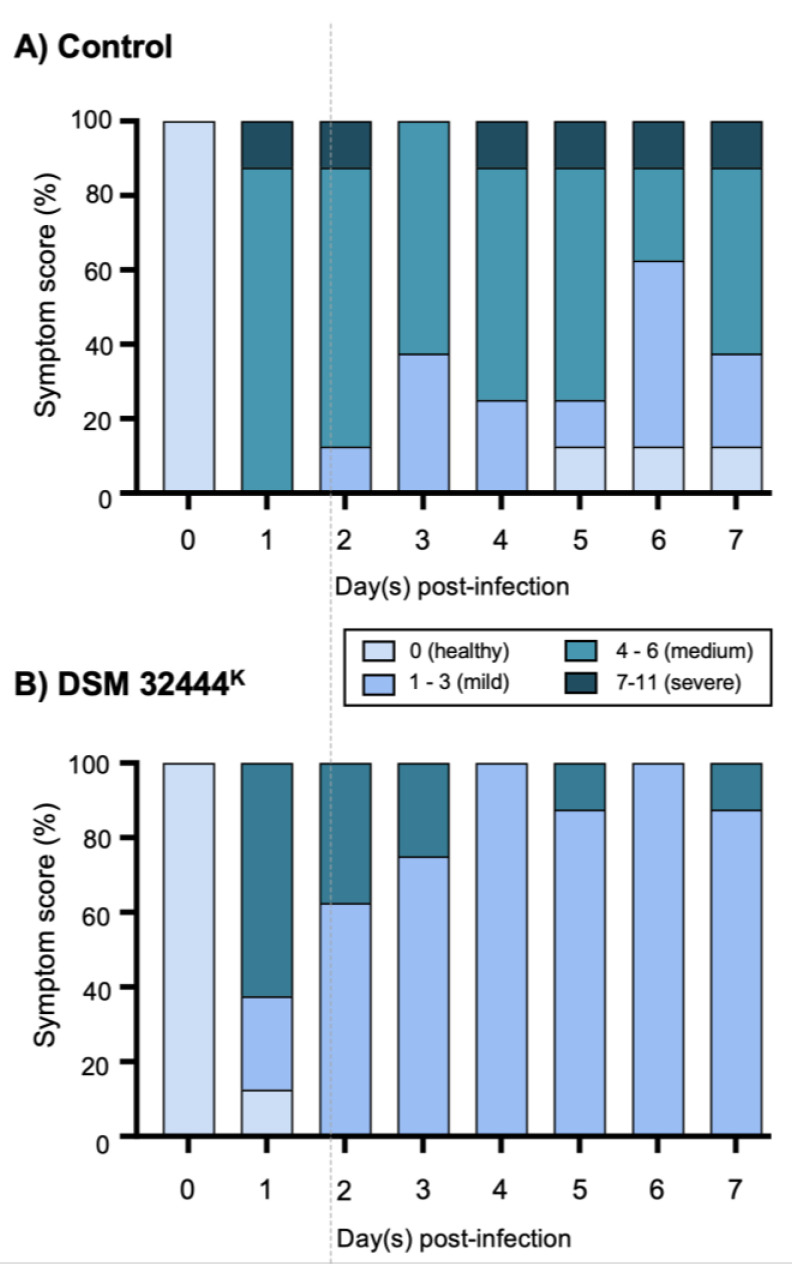
Clinical signs, change in body temperature, and weight loss exhibited by ferrets treated with DSM 32444^K^ (panel **B**) or mock-treated (panel **A**) with PBS following infection with H7N9 AIV. Ferrets were treated with 0.5 mL of DSM 32444^K^ or PBS on four separate occasions, 7 days apart, followed by infection with H7N9 AIV 7 days after the last treatment. Individual values were plotted per animal, and lines indicate the mean values per group. All ferrets were scored for clinical signs daily following infection according to the clinical score system presented in Appendix A. The cumulative daily clinical scores were represented graphically.

**Figure 2 vaccines-10-01559-f002:**
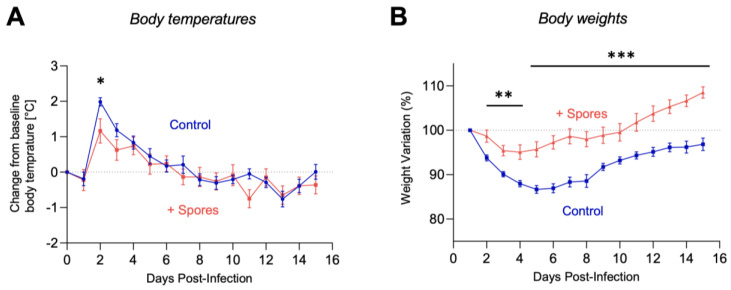
Change in body temperature and weight loss exhibited by ferrets treated with DSM 32444^K^ or mock treated with PBS following infection with H7N9 AIV. Ferrets were treated with 0.5 mL of DSM 32444^K^ (red line) or PBS (blue line) on four separate occasions, 7 days apart, followed by infection with H7N9 AIV 7 days after the last treatment. Individual values were plotted per animal, and lines indicate the mean values per group. Body temperatures and weights taken 1 day prior to infection were used to configure a baseline, and then, temperatures (panel **A**) and weights (panel **B**) were taken daily following infection with H7N9 AIV. Statistical significance was determined using a two-tailed Mann-Whitney U test; panel A, * *p* < 0.05; panel B, ** *p* < 0.01, *** *p* < 0.001.

**Figure 3 vaccines-10-01559-f003:**
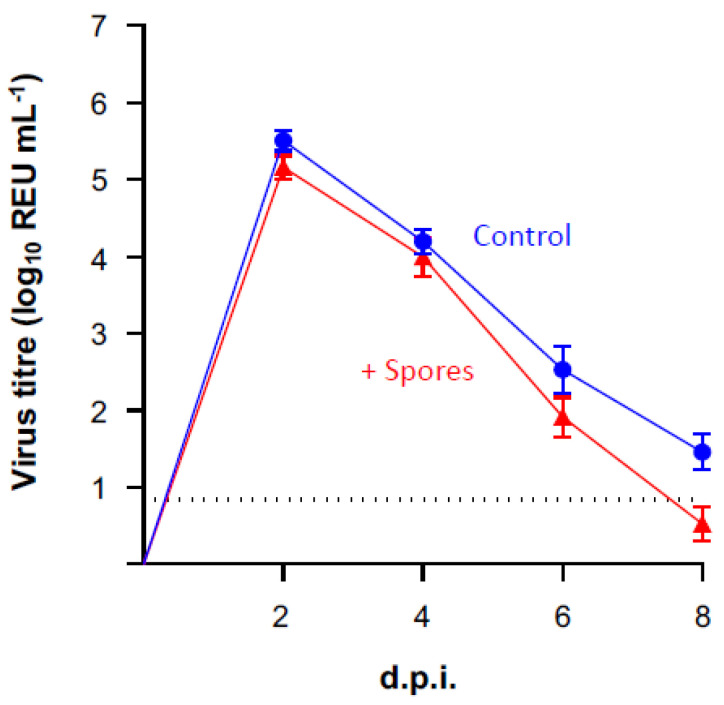
Viral shedding exhibited by ferrets treated with DSM 32444^K^ or mock-treated with PBS following infection with H7N9 AIV. Viral RNA was quantified in RNA extracted from nasal wash samples using an influenza A rRT-qPCR. Relative equivalency units (REUs) were calculated and displayed based on Cq values obtained extrapolated from a standard curve of a known titre of A/Anhui/1/13 (H7N9).

## Data Availability

All data generated or analyzed during this study are included in this published article (and its Appendix A) and is available by request.

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
