# Peer review of "Intranasal Treatment of Ferrets with Inert Bacterial Spores Reduces Disease Caused by a Challenging H7N9 Avian Influenza Virus"

_vaccines, 2022, doi:10.3390/vaccines10091559_

Round 1

Reviewer 1 Report

Comments:

1.      In this study, the authors stated that blood was obtained both prior to infection via venipuncture and post-infection via cardiac puncture under terminal anesthesia. However, only the antibody detection for post-infection was presented in the results.

2.      It is highly recommended to pay attention to the animal ethics of this study. After challenge with A/Anhui/1/13 [H7N9], most ferrets exhibited weight loss with a 20% reduction in weight from baseline data between 1 and 5 dpi, while, the control treated groups exhibited significantly greater weight loss compared to 142 DSM 32444K-treated ferrets, with more than 50% reduction in weight from baseline data. Dose more than 50% reduction in weight meet ethical requirements for animal experiments? By the way, the ferrets in control group exhibited weight loss with a less than 3.7% reduction in weight from baseline data after infection with A/Anhui/1/13 [H7N9](PMID: 23868922).

Author Response

Thank you for the informative and useful reviews. I have provided comments to the points raised in bold type below.

The use of spores as an immunostimulant and vaccine vehicle has been well studied. The importance of this work lies mainly in the animal model and the highly pathogenic virus. However, this work would be stronger if was assessed the spore-based vaccine against H7N9. 

With regard to the last sentence of the paragraph above. We are unsure of the meaning and whether there might be some misunderstanding of our approach. If so we apologise and hope to clarify.

We have previously published a paper (Song et al 2011) that is cited in the paper here that does indeed refer to a vaccine using spores. This vaccine used spores adsorbed with influenza immunogens (intact virions). The immunisation of mice with adsorbed spores produced adaptive immunity and protection. The Song et al paper also refers to the observation that heat inactivated spores alone (ie, no immunogen) also induced protection. We showed that this was by innate immunity. Thus, there were 2 methods of prophylaxis described in the paper, vaccination and immunomodulation/innate immunity. It was this phenomenon that prompted the paper submitted here. Indeed, one criticism we have always had was that the mouse model was insufficient and a second, more meaningful model should be used. It was for this reason that we used here ferrets. We focused only on using killed spores and innate immunity and considered the use of adsorbed spores as being unlikely to have long term utility in influenza prophylaxis since vaccines are already in use and 50 years of effort has led to an organised programme of new seasonal flu vaccines. However, we did believe that killed spores might have utility as an adjunct to existing flu vaccines. Since they are non-GMO, and can be stored at ambient temp it is possible to consider these as useful adjuncts to existing flu vaccination programmes.

This work would be more complete if including the level of antibodies in two groups, and see if it increases in the immunostimulated animals, even if the number of animals is small.

Again, as mentioned above we are not using an approach that generates adaptive immunity and we are not delivering any influenza immunogens so it would not be possible to examine responses to influenza antigens other than to verify whether infection had occurred post challenge (which we did and is a cited at the end of results).

The discussion is very brief. Please, this should be extended, for example, in regard to TLRs described as targets of B. subtilis spores. In addition, you should include more discussion about mechanisms implied in innate response implied after spore stimulation.

Please note that we have summarised what is known about spores and their interaction with TLRs and subsequent effects on NK cells and maturation on DCs in the introduction. This is based on mouse studies. I have added more detail in the discussion however.

Rev 1

   In this study, the authors stated that blood was obtained both prior to infection via venipuncture and post-infection via cardiac puncture under terminal anesthesia. However, only the antibody detection for post-infection was presented in the results. 

We were only interested in measuring antibodies to confirm infection (+/-) between treated an untreated groups

We were not interested in looking at antibody levels over time since we were not immunizing with immunogens

We also state in methods that all animals were checked for absence of influenza (by PCR) prior to study start

In sum, we believe our approach was most suitable for the study.

  1. It is highly recommended to pay attention to the animal ethics of this study. After challenge with A/Anhui/1/13 [H7N9], most ferrets exhibited weight loss with a 20% reduction in weight from baseline data between 1 and 5 dpi, while, the control treated groups exhibited significantly greater weight loss compared to 142 DSM 32444K-treated ferrets, with more than 50% reduction in weight from baseline data. Dose more than 50% reduction in weight meet ethical requirements for animal experiments? By the way, the ferrets in control group exhibited weight loss with a less than 3.7% reduction in weight from baseline data after infection with A/Anhui/1/13 [H7N9](PMID: 23868922).

We apologise. We have made a mistake on the figure (Fig 2B)

The y axis had three data points 110%, 100% and 50%. This should be 110%, 100% and 90%

I have now corrected the figure and also expanded it to 14 days. From that you will see that the max weight loss was <15% and is in accordance with our project licence limits of 20%. We have also expanded Fig 2A top 14 days

We have now made the shedding figure as a separate figure (Fig 3)

Reviewer 2 Report

Protection of ferrets against avian influenza H7N9 challenge following intranasal dosing of inert bacterial spores

By HA Hong et al (Corresponding author: SM Cutting)

Submitted to Vaccines (Editorial No: vaccines-1856236)

General Comments

Avian influenza viruses (AIV) can become a zoonotic threat for humans and potentially the cause of epidemics. Here the ferret model of influenza virus infection is used to show that pretreatment of  animals with killed (inert) spores of B subtilis (by nasal inoculation), followed by challenge with the AIV A/Anhui/1/13 [H7N9] (by nasal inoculation), resulted in milder disease (as measured by weight loss, body temperature and a clinical score) than observed in mock-treated animals. Virus shedding into the nasal cavity, however, was not reduced, indicating that productive infection had taken place in the spore-treated animals. It is concluded that inert bacterial spores could become a candidate for prophylaxis or treatment of zoonotic avian or human influenza.

From an earlier publication [no 3] in which mice are used as animal influenza infection model, it was deduced that inert spore treatment may activate pathways of innate immune responses (IIR) and that this is the likely mechanism of action, thought to be operative also in ferrets. However, no details on IIRs in the ferrets are provided. It had previously been observed [3] that inert spores can also act as adjuvants for influenza vaccination and that spores can bind H5N1 AIVs. Clarification is requested on a number of points (dosing of pretreatment with inert spores, dose of challenging virus, interaction of H7N9 virus with inert spores, etc). With the available data, the proposal to use inert bacterial spores for prophylaxis of influenza is not fully convincing.

Specific Comments

Line

2          Reconsider title, e.g. ‘Intranasal treatment of ferrets with inert bacterial spores reduces disease caused by a challenging H7N9 avian influenza virus’ , or similar.

15        Consider phrasing: … ‘universal’ prophylactic approaches are under development.

20        … Virus shedding was slightly reduced but not prevented.

51        Guinea pigs and Syrian hamsters have been used as an influenza infection and transmission model.                                                                                                                       Lowen AC, et al. The guinea pig as a transmission model for human influenza viruses. Proc Natl Acad Sci U S A. 2006 Jun 27;103(26):9988-92. [and various follow up papers]

Fan S, et al. Influenza Viruses Suitable for Studies in Syrian Hamsters. Viruses. 2022 Jul 26;14(8):1629.

58        … We show that intranasal application of inert spores led to …

65        More details should be reported on the morphology of autoclaved spores. The test used to prove spore inactivation should be described.

88        The titer of the challenging virus 107 TCID50/ml, appears to be very high. A statement should be made on the disease producing dose 50 percent of the challenging virus for the ferret model.

157      Fig. 2, Legend. The colour code of the curves should be explained.

177      Since the data do not show a statistically significant difference, the sentence should be rephrased.

179      Some data should be provided on the infectivity of the shed challenge virus.

183      Since pretreatment with inert spores of ferrets does not seem to reduce viral shedding significantly and is thus unlikely to reduce the potential for transmission, this weakens the argument of using inert spores for prophylaxis.  

203      More detailed information is required about the possible mechanism of AIV H7N9 disease reduction in the ferret model. How does this virus interact with inert spores? Furthermore, have spores been treated with neuraminidase (RDE) or periodate before mixing with virus? Have alternative dosing schedules for intranasal spore application been considered? Have lower infectivity doses of the challenging virus been considered? [The dose of challenging virus used in the mouse experiments [3] appears to have been much lower. The data on virus shedding suggest that there may have been an excess of challenging virus.]

Author Response

Thank you for the informative and useful reviews. I have provided comments to the points raised in bold type below.

The use of spores as an immunostimulant and vaccine vehicle has been well studied. The importance of this work lies mainly in the animal model and the highly pathogenic virus. However, this work would be stronger if was assessed the spore-based vaccine against H7N9. 

With regard to the last sentence of the paragraph above. We are unsure of the meaning and whether there might be some misunderstanding of our approach. If so we apologise and hope to clarify.

We have previously published a paper (Song et al 2011) that is cited in the paper here that does indeed refer to a vaccine using spores. This vaccine used spores adsorbed with influenza immunogens (intact virions). The immunisation of mice with adsorbed spores produced adaptive immunity and protection. The Song et al paper also refers to the observation that heat inactivated spores alone (ie, no immunogen) also induced protection. We showed that this was by innate immunity. Thus, there were 2 methods of prophylaxis described in the paper, vaccination and immunomodulation/innate immunity. It was this phenomenon that prompted the paper submitted here. Indeed, one criticism we have always had was that the mouse model was insufficient and a second, more meaningful model should be used. It was for this reason that we used here ferrets. We focused only on using killed spores and innate immunity and considered the use of adsorbed spores as being unlikely to have long term utility in influenza prophylaxis since vaccines are already in use and 50 years of effort has led to an organised programme of new seasonal flu vaccines. However, we did believe that killed spores might have utility as an adjunct to existing flu vaccines. Since they are non-GMO, and can be stored at ambient temp it is possible to consider these as useful adjuncts to existing flu vaccination programmes.

This work would be more complete if including the level of antibodies in two groups, and see if it increases in the immunostimulated animals, even if the number of animals is small.

Again, as mentioned above we are not using an approach that generates adaptive immunity and we are not delivering any influenza immunogens so it would not be possible to examine responses to influenza antigens other than to verify whether infection had occurred post challenge (which we did and is a cited at the end of results).

The discussion is very brief. Please, this should be extended, for example, in regard to TLRs described as targets of B. subtilis spores. In addition, you should include more discussion about mechanisms implied in innate response implied after spore stimulation.

Please note that we have summarised what is known about spores and their interaction with TLRs and subsequent effects on NK cells and maturation on DCs in the introduction. This is based on mouse studies. I have added more detail in the discussion however.

General Comments

Avian influenza viruses (AIV) can become a zoonotic threat for humans and potentially the cause of epidemics. Here the ferret model of influenza virus infection is used to show that pretreatment of  animals with killed (inert) spores of B subtilis (by nasal inoculation), followed by challenge with the AIV A/Anhui/1/13 [H7N9] (by nasal inoculation), resulted in milder disease (as measured by weight loss, body temperature and a clinical score) than observed in mock-treated animals. Virus shedding into the nasal cavity, however, was not reduced, indicating that productive infection had taken place in the spore-treated animals. It is concluded that inert bacterial spores could become a candidate for prophylaxis or treatment of zoonotic avian or human influenza.

From an earlier publication [no 3] in which mice are used as animal influenza infection model, it was deduced that inert spore treatment may activate pathways of innate immune responses (IIR) and that this is the likely mechanism of action, thought to be operative also in ferrets. However, no details on IIRs in the ferrets are provided. It had previously been observed [3] that inert spores can also act as adjuvants for influenza vaccination and that spores can bind H5N1 AIVs. Clarification is requested on a number of points (dosing of pretreatment with inert spores, dose of challenging virus, interaction of H7N9 virus with inert spores, etc). With the available data, the proposal to use inert bacterial spores for prophylaxis of influenza is not fully convincing.

I have added more details on what is known about innate immunity in ferrets with references

I have also tried to clarify that we are not using a true vaccine approach and we are NOT delivering an influenza  immunogen

Specific Comments

Line 

2          Reconsider title, e.g. ‘Intranasal treatment of ferrets with inert bacterial spores reduces disease caused by a challenging H7N9 avian influenza virus’ , or similar.

 Yes, the suggested title is better and we have modified

15        Consider phrasing: … ‘universal’ prophylactic approaches are under development.

rephrased to.... Vaccination has failed to halt the evolution of this virus and auxiliary prophylactic approaches are under development.

20        … Virus shedding was slightly reduced but not prevented.

Corrected

51        Guinea pigs and Syrian hamsters have been used as an influenza infection and transmission model.                                                                                                                       Lowen AC, et al. The guinea pig as a transmission model for human influenza viruses. Proc Natl Acad Sci U S A. 2006 Jun 27;103(26):9988-92. [and various follow up papers]

Fan S, et al. Influenza Viruses Suitable for Studies in Syrian Hamsters. Viruses. 2022 Jul 26;14(8):1629. 

I have revised the sentence which is improved, and also added refere3nces, thank you

58        … We show that intranasal application of inert spores led to …

Corrected

65        More details should be reported on the morphology of autoclaved spores. The test used to prove spore inactivation should be described.

I have added detail and references

88        The titer of the challenging virus 107 TCID50/ml, appears to be very high. A statement should be made on the disease producing dose 50 percent of the challenging virus for the ferret model.

The challenge titer was pre-validated by APHA (who conducted the study). The challenge dose is with published data and for example de Jonge (de Jonge, J.; Isakova-Sivak, I.; van Dijken, H.; Spijkers, S.; Mouthaan, J.; de Jong, R.; Smolonogina, T.; Roholl, P.; Rudenko, L. H7N9 Live Attenuated Influenza Vaccine Is Highly Immunogenic, Prevents Virus Replication, and Protects Against Severe Bronchopneumonia in Ferrets. Mol Ther 2016, 24, 991-1002, doi:10.1038/mt.2016.23.) used the same dose in a recent ferret study. I have cited de Jonge in the paper.

157      Fig. 2, Legend. The colour code of the curves should be explained.

apologies, corrected

177      Since the data do not show a statistically significant difference, the sentence should be rephrased.

179      Some data should be provided on the infectivity of the shed challenge virus.

this was not tested but I have made modified this as follows

"Infectivity of shed virus was not determined and although rRT-qPCR is highly sensitive and was used here to detect viral RNA it is unable to provide insight into the titers of infectious virus present within the sample for which egg isolation is traditionally used."

183      Since pretreatment with inert spores of ferrets does not seem to reduce viral shedding significantly and is thus unlikely to reduce the potential for transmission, this weakens the argument of using inert spores for prophylaxis.  

this section has been rewritten and toned down

203      More detailed information is required about the possible mechanism of AIV H7N9 disease reduction in the ferret model. How does this virus interact with inert spores? Furthermore, have spores been treated with neuraminidase (RDE) or periodate before mixing with virus? Have alternative dosing schedules for intranasal spore application been considered? Have lower infectivity doses of the challenging virus been considered? [The dose of challenging virus used in the mouse experiments [3] appears to have been much lower. The data on virus shedding suggest that there may have been an excess of challenging virus.]

no, spores were not pretreated with neuramindase but they were also not mixed with virus

the virus was dosed later at the challenge day (7 days after the last dose of spores)

note then that the virus was never bound or adsorbed to the spores

the other comments regarding dosing schedules, regimen, dose etc we have considered and they are mentioned in the rewritten section (page 9)

Round 2

Reviewer 1 Report

Major comments:

1.      Line 41-43, the conclusion, most likely as an adjunct to existing vaccination programs could provide a useful approach for improving influenza prophylaxis in both humans and animals, is not suitable for this study. The experiment designs did not include the combination use of the test vaccine and the existing programs.

2.      Line 114-116, the authors stated that blood was obtained both prior to infection via venipuncture, immediately after this period (Day 0) animals were confirmed free of current influenza infection with HAI titers 1/5 and absence of viral RNA in nasal wash samples. And this antibody detection is important to know the antibody background of the experimental ferrets. Although the authors had detected the H7N9 virus prior to infection, the results could present the antibody data before the experiments.

3.      Line 127-128, the day presentation is confused in the expression. Such as, on Day 42 (12 days post-infection, d.p.i.) or at a defined endpoint animals were euthanized and blood taken for serological analysis. And the data in the Supp. Fig. 1, the data were collected during the 28 days post 1st dose. The special day is the same day or not at the end of the experiments. Please confirm it.

4.      The revised fig 2 and figure 3 can’t be found in the revised manuscript.

5.      In legends of the Supp. Fig. 1, what’s the lines of different color represent? Which groups? Were the data of the Supp. Fig. 1 consistent with the data of figure 2?

Author Response

  1. Line 41-43, the conclusion, most likely as an adjunct to existing vaccination programs could provide a useful approach for improving influenza prophylaxis in both humans and animals, is not suitable for this study. The experiment designs did not include the combination use of the test vaccine and the existing programs.

Yes, agreed, the tone is too optimistic. I have changed to

"Taken together nasal dosing using heat-stable spores could provide a useful approach for influenza prophylaxis in both humans and animals."

  1. Line 114-116, the authors stated that blood was obtained both prior to infection via venipuncture, immediately after this period (Day 0) animals were confirmed free of current influenza infection with HAI titers ≤1/5 and absence of viral RNA in nasal wash samples. And this antibody detection is important to know the antibody background of the experimental ferrets. Although the authors had detected the H7N9 virus prior to infection, the results could present the antibody data before the experiments. 

I have may have made an error. having rechecked what we have written and discussion with APHA the antibody titres were measured and this is actually already stated on line 114 with titres of <1/5 of the original version of the ms

I have modified the sentence to state considered free of infection rather than confirmed free of infection

  1. Line 127-128, the day presentation is confused in the expression. Such as, on Day 42 (12 days post-infection, d.p.i.) or at a defined endpoint animals were euthanized and blood taken for serological analysis. And the data in the Supp. Fig. 1, the data were collected during the 28 days post 1st dose. The special day is the same day or not at the end of the experiments. Please confirm it. 

lines 127-128 revised to Twelve days post-infection, (d.p.i.) or at a defined endpoint animals were euthanized and blood taken for serological analysis.

The supp fig 1 has also been revised

  1. The revised fig 2 and figure 3 can’t be found in the revised manuscript.

The figures have been inserted in the manuscript

  1. In legends of the Supp. Fig. 1, what’s the lines of different color represent? Which groups? Were the data of the Supp. Fig. 1 consistent with the data of figure 2?

A key to the legend has been added showing groups. Supp. Fig. 1 is clinical signs (weights and temperatures) before viral challenge. Yes, the data was consistent between preinfection and post infection.

Reviewer 2 Report

Intranasal treatment of ferrets with inert bacterial spores reduces disease caused by a challenging H7N9 avian influenza A virus

By HA Hong et al (Corresponding author: SM Cutting)

Submitted to Vaccines (Editorial No. vaccines-1856236R1)

General Comments

This is the revised version (R1) of a manuscript the original of which has been studied and commented by this reviewer. The authors have considered the comments and suggestions very carefully, and the R1 manuscript has vastly improved. The reviewer has only relatively minor additional comments.

Specific Comments

Line

27           Consider phrasing: … and next generation prophylactic approaches are under development.

32           … spores (…) every 7 days for 4 weeks.

33           … challenged with the avian H7N9 influenza A virus.

60           … shown to be innate immunity…

63           … Secondly, they have adjuvant properties…

68           … Seasonal influenza A and B viruses…   cases/year …

71           … avian and swine influenza…

73           … potentially increase the risk…

95           … with no bacteria growing, meeting the required standard.

131         … for clinical symptoms using a standardized scoring system (Table S1) …

141         … at 1000 g for … minutes… [Please insert]

147         Add a statement of how inactivation by beta-propiolactone was tested.

158         … calculcated based on Cq (quantification cycle) values…

167         … Figures S1 and 2 …

170         … For the remaining animals…

182         … administration of DSM32444k spores…

188         and line 190. Read: Fig. 3  .

200         to line 207. Try to condense and make less speculative.

220         … spores of B. subtilis leads to…

221         … influenza virus infection.

222         The reduction of clinical symptoms…

224         … which measured only…

227         … protection from disease…

232         … most typically Toll-like receptors…

237         … Inert spores than add to the number of…

261         … [35], and humans and animals are exposed…

262         … with the exposure estimated to be 104-105 spores/day.

274         Considering that pandemics are likely to occur in the future it seems…

277         Since the underlying mechanism of spore application is likely innate immunity, it will be of interest…

438         … according to the clinical score system presented in Suppl. Table S1.

446         Please clarify the timing.

456         The sentence ‘Dotted lines… ..7.012’ should be omitted.

Author Response

Thank you 

these were very helpful

all changes made as listed below

Specific Comments 

Line 

27           Consider phrasing: … and next generation prophylactic approaches are under development.

thank you, change made

32           … spores (…) every 7 days for 4 weeks.

change made

33           … challenged with the avian H7N9 influenza A virus.

change made

60           … shown to be innate immunity…

change made

63           … Secondly, they have adjuvant properties…

change made

68           … Seasonal influenza A and B viruses…   cases/year …

change made

71           … avian and swine influenza…

change made

73           … potentially increase the risk…

change made

95           … with no bacteria growing, meeting the required standard.

change made

131         … for clinical symptoms using a standardized scoring system (Table S1) …

change made

141         … at 1000 g for … minutes… [Please insert]

change made

147         Add a statement of how inactivation by beta-propiolactone was tested.

I have added the reference for the method

158         … calculcated based on Cq (quantification cycle) values…

change made

167         … Figures S1 and 2 …

change made

170         … For the remaining animals…

change made

182         … administration of DSM32444k spores…

change made

188         and line 190. Read: Fig. 3  .

thank you, corrected!!

200         to line 207. Try to condense and make less speculative.[1]

I have revised

220         … spores of B. subtilis leads to…

change made

221         … influenza virus infection.

change made

222         The reduction of clinical symptoms…

change made

224         … which measured only…

change made

227         … protection from disease…

change made

232         … most typically Toll-like receptors…

change made

237         … Inert spores than add to the number of…

change made

261         … [35], and humans and animals are exposed…

change made

262         … with the exposure estimated to be 104-105 spores/day.

change made

274         Considering that pandemics are likely to occur in the future it seems…

change made

277         Since the underlying mechanism of spore application is likely innate immunity, it will be of interest…

change made

438         … according to the clinical score system presented in Suppl. Table S1.

change made

446         Please clarify the timing.

I have revised the sentence to make clearer

456         The sentence ‘Dotted lines… ..7.012’ should be omitted.

change made

Round 3

Reviewer 1 Report

1.      Line 128-129, the author stated that twelve days post-infection, (d.p.i.) or at a defined endpoint animals were euthanized and blood taken for serological analysis, which is not consistent with the statement that serum was collected from all ferrets at 14 dpi on line 216. While, in the results, the related data at 15 dpi were showed in fig 2. It is difficult to understand the method and the results.

2.      Line 195, there are panel A and panel B in Fig 2. Where is the Fig 2C?The authors should carefully check these kinds of low-level mistakes throughout the manuscript.

Author Response

Point 1

Yes, error relating dpi

this has been corrected to 15 dpi in the manuscript and highlighted

Point 2

we have amended (highlighted), apologies

Round 4

Reviewer 1 Report

Most of the comments have been addressed in the revised manuscript. Some minor comments should be paid attention before its publication in Vaccines:

Line 128-129, the author used d.p.i. (the days post-infection) in the expression, while in other parts of the manuscript, they used dpi. Please uniform the usage.

Line 88 and 120, the same problem includes hemagglutination inhibition titers (HITs) and HAI titers. Please confirm and uniform the usage.

Author Response

Both of these comments have been addressed with d.p.i. and HIT being used throughout